# Detection of Pesticides in Water through an Electronic Tongue and Data Processing Methods

Jeniffer Katerine Carrillo Gómez *, Yuliana Alexandra Nieto Puentes, Dayan Diomedes Cárdenas Niño and Cristhian Manuel Durán Acevedo *

Multisensor System and Pattern Recognition Research Group (GISM), Engenering and Architecture Faculty, Universidad de Pamplona, Pamplona 543050, Colombia
* Correspondence: jeniffer.carrillo@unipamplona.edu.co (J.K.C.G.); cmduran@unipamplona.edu.co (C.M.D.A.); Tel.: +57-312-4092247 (J.K.C.G.); +57-311-2135846 (C.M.D.A.)

**Abstract:** This study highlights the implementation of an electronic tongue composed of carbon screen-printed electrodes, which were used to discriminate and classify pesticides, such as Curathane, Numetrin, and Nativo in water. Therefore, to verify the capacity and performance of the sensory system, solutions of each of the pesticides at a concentration of 10 ppm were prepared in the laboratory and compared with distilled water. Furthermore, to evaluate the minimum detection limit of the electronic tongue, solutions were prepared at different concentrations: 0.02, 0.04, 0.06, 0.08, 0.1, 0.15, 0.2, and 0.25 ppm, respectively. The analysis and classification of the different categories and concentrations were obtained from the use of pattern recognition and automatic learning methods, such as principal component analysis (PCA), linear discriminant analysis (LDA), support vector machine (SVM), k-nearest neighbors (kNN), and naïve Bayes, during this process; the techniques accomplished more than 90% accuracy in pesticide concentrations. Finally, a 100% success rate in classifying the compound types was completely achieved.

**Keywords:** electronic tongue; pesticides; screen-printed electrodes; pattern recognition; machine learning





## 1. Introduction

Population growth is increasing at an accelerated rate; it has significant implications for the agricultural sector due to the increase in demand and food supply worldwide. It is estimated that by the year 2050, the world agricultural production will have to double its manufacture in order to satisfy this type of growing demand [1,2]. In addition, according to the need and urgency to achieve greater production and quality in food that complies with the current food safety regulations of each country, it is necessary to use pesticides as a frequently used tool to protect crops from possible losses of yield and reduction in the product quality and to guarantee high profits for the farmers by providing reliable supplies of agricultural products at affordable prices for consumers [3–5].

Pesticides are chemical complexes that act to prevent, combat, repel, control, or destroy pests. On the other hand, considering the type of disease or pest they treat, they are classified as herbicides, insecticides, rodenticides, and fungicides [6–8]. Herbicides and insecticides are the most used pesticides, dominating 47.5 and 29.5% of total pesticide consumption [9–11]. It is noteworthy that about a third of agricultural products are produced based on the application of pesticides. Without pesticides, there would be a 78% loss in fruit production, a 54% loss in vegetable production, and a 32% loss in cereal production [12,13]. Thus, pesticides have significantly contributed to relieving hunger and providing access to an abundant supply of high-quality food [14,15]. Nevertheless, it is difficult to obtain precise estimates of the agricultural losses caused by pests at national and international levels because the damage originated by these organisms depends on a number of factors related to environmental conditions, the plant species being cultivated, the farmers' socioeconomic conditions, and the level of technology implemented [16,17].

Although the application of pesticides provides a variety of benefits, such as improving the quality of food and growing the amount of food production by reducing problems related to pests of crop plants, as mentioned above, pesticides are considered persistent organic pollutants (POPs), which are a group of pollutants that generate global concern due to their bioaccumulation properties, high toxicity, and ubiquitous exposure to human beings and wildlife [18,19]. The reckless use of pesticides and other persistent organic pollutants in agricultural soils will have devastating repercussions in the future because many of the pesticides are associated with environmental and health problems due to their bioaccumulation properties and high toxicity [14,20]. Within a human or animal body, pesticides can be metabolized, excreted, stored, or bioaccumulate, which has generated numerous adverse health effects, such as dermatological [21], gastrointestinal [22,23], neurological [24–27], carcinogenic [28–30], respiratory [31–34], reproductive [35], and endocrine conditions [36–39]. Additionally, high pesticide exposure can result in hospitalization and death [40].

Regarding environmental impacts, pesticides can contaminate the soil, subsoil, surface, and groundwater from the grass and other flora; it can also remain in crops because there are pesticides that do not break down easily and accumulate (through uptake by the plant and soil particles adhering to the plant surface) and will eventually enter the food chain, posing a threat to humans, as residues of these can be found in a wide variety of everyday food and drinks [12,41–43].

For instance, the occurrence of pesticides in water derives from agricultural field runoff and industrial wastewater [44]. Despite the fact that the soil matrix serves as a storage compartment for pesticides due to the high affinity of agrochemicals with the soil, surface water resources, such as streams, estuaries, and lakes, as well as groundwater, are susceptible to contamination by pesticides due to the close interconnection of the soil with water bodies [8,45]. The low concentration of pesticides accumulated in water can increase along the food chain and enter into aquatic organisms that are dangerous for human consumption [9,14]. The presence of pesticides in water is regulated internationally through different regulatory frameworks [46,47]; in the European Union, we can find several directives, including Directive 2006/118/EC on Groundwater [48], the Drinking Water Directive 98/83/EC [49], the Water Framework Directive [49], which has subsequently been amended several times [50,51], and the new EU directive 2020/2184 on water quality intended for human utilization [52].

In the United States (USA), the Environmental Protection Agency (EPA) regulates pesticides at the national level in accordance with the Federal Insecticide, Fungicide, and Rodenticide Act (FIFRA) and other laws [53]. In Colombia, the regulation of pesticides is found in Decree 475/1998, "Normas Técnicas de Calidad del Agua Potable" [54]. The first two regulations agree that the maximum concentration for each pesticide must be 0.0001 and 0.0005 mg/L for the sum of all the pesticides present in a sample; however, in the case of Colombia, the maximum permissible limit depends on the toxicity classification made by the Ministry of Health and contemplated in Decree 1843/1991 [55].

The highest admissible concentration for pesticides with toxicological category I (highly toxic) is 0.0001 mg/L, toxicological categories II and III (medium and moderately toxic) is 0.001 mg/L, toxicological category IV (low toxicity) value is 0.01 mg/L, and the total sum of pesticide concentrations and other substances whose maximum admissible individual value is 0.0001 mg/L with a maximum of 0.001 mg/L; in no case may the individual values be exceeded [54].

One procedure to mitigate pesticide contamination is through an effective monitoring program, which includes sampling and chemical analysis [9,56]. Monitoring is also essential to assess the effectiveness of mitigation measures (for example, bans and restrictions on use, safe handling procedures, and integrated pest management) aimed at curbing pesticide contamination [15,18,57].

Most of the methods suggested by the Environmental Protection Agency (EPA) for the detection of pesticides in water are based on analytical techniques, such as high-

performance liquid chromatography (HPLC) [58,59], gas chromatography (GC) [44,60–62], micellar electrokinetic chromatography (MEKC), enzyme-linked immunosorbent assays (ELISA), and gas–liquid chromatography coupled with mass spectrometry (GC/MS, LC-MS) [63,64]. Despite the high precision and sensitivity accomplished in these methods, most of them require sample preparation, expensive equipment, qualified personnel, laboratories, and time, and they are not portable which makes their use highly difficult [64].

Nowadays, there is a need to create and implement reliable, fast-response, and low-cost devices that allow the monitoring of contaminants in different matrices (water, soil, and so on), as in the case of the present study on the detection of pesticides in water.

Electronic tongues (ETs) are promising analytical devices that can be used for the identification, classification, or quantification of chemical and/or biological families in complex matrices, which can deliver fast and accurate information in a cost-effective manner. Their high sensitivity for measurements in complex liquid media, minimal power requirement, low cost, easy integration with other systems, portability, and ease of use are some of their many advantages [65,66]. ETs are defined as a set of non-specific chemical electrodes with partial sensitivity (cross-sensitivity) to different components, capable of analyzing complex liquids [67,68]. The electrodes produce signals that are not necessarily specific to any particular species in the liquid, but a signal pattern is generated that can be related to certain features or qualities of the sample using appropriate software [69]. Electronic tongues are typically applied to give qualitative responses about the studied sample and predict the concentration of individual species in the sample in most cases [69,70]. Numerous chemical and biochemical (enzymatic) electrodes have been implemented to develop ET, exploring different detection techniques, such as electrochemical, potentiometric, voltametric, amperometric, impedimetric, and conductometric, which are the most widely used [71].

ETs have been implemented in many research fields, such as pharmacy [72–74] and biotechnology [75]; nonetheless, these devices have been mainly focused on food quality analysis [69,76–79] and environmental monitoring, where the electrode involves monitoring the quality of soil, water, and air [65,80–83]. Consequently, environmental monitoring is mainly directed at the detection of toxic compounds and contaminants (whether chemical or biological/microbiological) [65,84–87]. In this order of ideas, we will focus on the most relevant applications of the detection of pesticides in water [65]. Cortina et al. detected the pesticides dichlorvos, carnofuran, and methylparaoxon in water using an enzyme biosensor-based ET [88,89]. Another important study was carried out by Alonso et al., who used an automatic ET for the online detection and quantification of three types of organophosphate and carbamate pesticides, applying enzymatic screen-printed biosensors as an alternative way to measure possible POPs, even with the challenge of identification in concentrations as low as 0.1 uM [90]. In 2017, Murilo et al. carried out the detection of traces of organophosphorus pesticides (OPs) and mixtures of these through an electronic tongue based on hybrid graphene nanocomposites, where the results obtained indicated that it can be used as a fast, simple, and low-cost alternative in the analysis of OP pesticide solutions below the concentration range allowed by the legislation of some countries [91,92].

The review reported by Sopanrao et al. on the nucleic acid-based aptamer sensors (single-stranded ribonucleic acids (ssRNA) or single-stranded deoxyribonucleic acids (ssDNA)) for the molecular diagnosis of toxic chemicals in food, water, human fluids, and the environment [93] reports a novel aptamer-based disposable, flexible, and screen-printed electrochemical sensor (Aptasensor) for the rapid detection of chlorpyrifos (CPF). The proposed sensor showed good CPF detection capability with a low detection limit (0.097 ng/mL) and a broad linear detection range (1 to $1 \times 10^5$ ng/mL). The sensors also obtained good selectivity against the most common interfering agents (dichlorvos, malathion, carbofuran, deltamethrin, and metamitron) [94].

In another study, a DNA Aptamer sensor was developed to detect fenitrothion (an insecticide belonging to the family of organophosphate pesticides) with a detection limit of 14 nM (3.88 ppb). Aptamer-based biosensors are often used as an alternative that provide a fast and easy way to recognize contamination from various sources. They also have

the advantage of detecting low concentrations that could not be detected by traditional chromatographic assays [95].

The objective of this study was to implement an electronic tongue with a set of screen-printed sensors for the detection and discrimination of three pesticides Curathane (cymoxanil and mancozeb), Nativo (tebuconazole and trifloxystrobin), and Numetrin (cypermethrin) present in water to demonstrate the potential of ET as an alternative technique for monitoring pesticides in water.

## 2. Materials and Methods

The following scheme describes the methodology used in this study (see Figure 1).

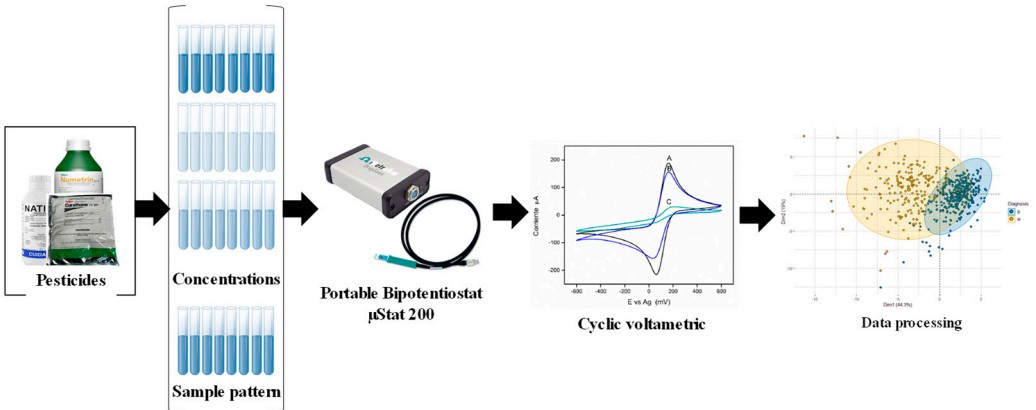

**Figure 1.** Overall scheme of the study based on an electronic tongue to detect pesticide types and concentrations in water.

### 2.1. Sample Preparation

The selection criteria of the pesticides carried out in this study was undertaken in collaboration with experts in this area and was conducted in Pamplona, Norte de Santander, Colombia. A small survey was applied to inquire about the type of pesticides used in the crops, the stage and frequency they were employed, the dosage or the approximate concentrations which the product was applied, and the possibility of purchase. According to this information, three pesticides were selected. (1) Curathane: it has cymoxanil (1[(EZ)-2-cyano-2-methoxyminoacetyl]-3-ethylurea) as an active ingredient and mancozeb, which is a complex of zinc manganese ethylenebisdithiocarbamate. (2) Nativo: it has tebuconazole and trifloxystrobin as active ingredients. (3) Numetrin: this pesticide is composed of cypermethrin as an active ingredient from the group of pyrethroids. The pesticides were purchased through a local supplier called "Septima Distribuciones Agropecuarias S.A.S" located in the city of Pamplona, Norte de Santander (Colombia). Curathane is distributed and manufactured by AgroSciences de Colombia S.A, with national registration number of ICA No. 2145; Nativo is distributed and manufactured by Bayer S.A company, where its national registration number of the Colombian Agricultural Institute (ICA) is 429; and Numetrin is distributed by Nufarm Colombia S.A, with the national registration number of ICA No. 2074, which is manufactured by Gharda Chemicals Limited from India. It should be clarified that the pesticides used in these assays were of high quality and widely used in Colombian regions.

The first part of this study consisted of evaluating the ability of ET to discriminate and classify samples of different types of pesticides and water without contaminants. To develop this procedure, a 10 ppm solution of the pesticides, such as Curathane, Numetrin, and Nativo as mentioned above, which were purchased at an agricultural products supply site, was prepared in the laboratory of the University of Pamplona (Colombia).

In the second part, a study was included to evaluate the detection limit of the multi-sensory system. Solutions of each pesticide were prepared at concentrations of 0.02, 0.04, 0.06, 0.08, 0.1, 0.15, 0.2, and 0.25 ppm.

### 2.2. Electronic Tongue

The sensory perception system used in this study was a portable, low-cost, linear Bi-potentiostat µStat 200 manufactured by the "DropSens" company, which works with a maximum measurable current of $\pm$ 200 µA and a potential of 12 VDC. It was implemented to carry out measurements using the most standard electrochemical techniques: amperometry and pulsed amperometry, linear and cyclic voltammetry, differential pulse voltammetry, and square wave voltammetry through software for the acquisition of electrochemical spectra called "DropView", for controlling the potentiostat device. The measurements were recorded, and the analysis of the results was performed [96].

In the sample analysis process, screen-printed carbon electrodes (C110) were used for the acquisition of the signals where the cyclic voltammetry (CV) method was applied to determine the current density that is generated by electrodeposition by locating the area of possible oxidation and reduction of the sample. For each category and concentration, eight repetitions were performed.

In the data acquisition, the following values were configured: Ebegin: 0; exploration start potential, Evtx1: $-1$; exploration inversion potential, Evtx2: +1; voltage with exploration stop, and the number of scans = 1. The potentiostat was set to auto mode using cyclic voltammetry analysis, and the acquisition time was performed for 2 min, at a scan rate of 0.05 V/s.

### 2.3. Data Processing

For data analysis, different pattern recognition methods and machine learning algorithms were used in this study. The principal component analysis (PCA) and linear discriminant analysis (LDA) algorithms were applied as multivariate analysis methods. On the other hand, for the sample classification, supervised learning methods, such as naïve Bayes, support vector machine (SVM), and the k-nearest neighbors (kNN) algorithm, were implemented.

#### 2.3.1. Principal Component Analysis (PCA)

PCA analysis is an unsupervised pattern recognition method and represents a successful use in multivariate data by calculating their coordinates which are called principal components (PCs), through the original data or input variables. These main components are formed through a linear combination of the input variables, and loadings are the contribution of the original variables [97]. The PCA method is very useful for reducing the high dimensionality of the data set while preserving the information through the variance. The first principal component minimizes the distance between the data set and maximizes the variance of the transformation points [98].

If X is a data matrix with m rows and n columns, each variable is a column, and each sample is a row. In addition, PCA decomposes X as the sum of r ti and pi, where r is the rank of the X matrix:

$$X = t1P^{T1} + t2P^{T2} + \ldots + tkP^{Tk} + \ldots + trP^{Tr} \tag{1}$$

where t are the PCs that indicate the relationship between measures and P the relationship between each variable.

#### 2.3.2. Linear Discriminant Analysis (LDA)

This is a linear model for classification and dimensionality reduction. It is a pattern recognition method that maximizes the distance between the means of two classes and minimizes the variation between each category [99]. LDA method processes the information of the measures where the categories or groups are considered in such a way that they have a normal distribution with similar dispersion. The objective of this analysis is to map the samples from an N-dimensional space to a linear one. The LDA algorithm creates a linear combination that produces differences between the desired values or classes. For all

samples of all categories, two measures are defined: (1) one is called within-class scatter matrix, given by:

$$S_w = \sum_{j=1}^{c} \sum_{i=1}^{N_j} (x_i^j - \mu_j)(x_i^j - \mu_j)^T \qquad (2)$$

where $x_i^j$ is the sample number of class j, $\mu j$ is the mean of class j, c is the number of classes, and $N_j$ is the number of samples in class j; and (2) it is called the dispersion matrix between categories, which is given by:

$$S_b = \sum_{i=1}^{N_j} (\mu_j - \mu)(\mu_j - \mu)^T \qquad (3)$$

where μ represents the mean of all classes. Therefore, the goal is to maximize class separation while minimizing the measures within the categories [100].

### 2.3.3. Naïve Bayes (Naïve Bayesian Classifier)

The naïve Bayes classifier is a probabilistic classification method based on Bayes' theorem with strong assumptions of independence between features.

This classifier has the main characteristic of having robust assumptions about the self-sufficiency of each condition. Compared to other classifier models, naïve Bayes performs appropriately. One of the benefits of this method is that a small amount of training data is needed to compute the estimated parameter used in the classification. The independent variable is the variation of a variable in a class that is designed to decide the classification and not the entire covariance matrix. The training stage and the classification stage are the naïve Bayes stages [101].

Therefore, the outset is to solve a classification task through the training data of a given number of classified objects t, which requires predicting the probability P (y|x) of which a new instance x = (x1, . . . , xa) belongs to some class y, where xi is the value of the attribute Xi, and furthermore, y ∈ {1, . . . , c} is the value of the class variable y [102].

### 2.3.4. Support Vector Machines (SVMs)

SVM is a classification method that projects a set of measurements onto an optimal "hyperplane" that serves to separate observations that belong to one class from another based on patterns of information about those observations called features. This hyperplane can be used to determine the most likely label for the unseen data [103].

Training through the SVM algorithm consists of determining the hyperplane to separate the data training belonging to a certain number of classes. Its location is defined by a small subset of vectors from the training set (T) called support vectors. One of the most widely implemented support machine algorithms for class separation is linear SVM, in which data are separated into a dimensional input space through the use of a hyperplane defined as a function:

$$f(x): w^T x + b = 0 \qquad (4)$$

where w is the normal vector of the hyperplane, w ∈ RD, and b/||w|| is the perpendicular distance between the hyperplane and the origin, b ∈ R. This hyperplane is located such that the distance between the closest vectors of the classes opposite the hyperplane is maximum [103]. Today, different kernels are used in SVM algorithms with the aim of taking data as input and transforming them into different forms; for example, linear, nonlinear, polynomial, radial basis function (RBF), sigmoid, among others.

### 2.3.5. k-Nearest Neighbors (kNN)

The nearest neighbor rule is one of the oldest methods of class reasoning. The decision-making is straightforward, where the sample to be tested is the same as the closest sample category. The kNN method has been widely used in data mining and machine learning

applications due to its simple implementation and good performance. The performance is remarkable in extensive data, where the error rate reaches approximately Bayesian optimization under very smooth conditions, which is considered one of the most used methods for solving classification problems. There are several distance functions to obtain similar observations; some of them can be calculated using the Manhattan, Mahalanobis, Minkowski distance and, among the best known, the Euclidean distance, which is defined as follows [104]:

$$D_E = \sqrt{\sum_{i=1}^{D} (x_i - y_i)^2} \tag{5}$$

where xi\ and yi (i = 1 ... N) are an attribute of two observations (samples). Therefore, through the selected nearest neighbors, two voting approaches are used to classify the test sample:

Major Vote: $y' = argmax_v \sum_{(x_i,y_i) \in D_z} \delta(v, y_i)$

Distance-weighted voting: $y' = argmax_v \sum_{(x_i,y_i) \in D_z W_i} \delta(v, y_i)$

Where $\delta$ ( ) is an indicator function, DZ is the set of nearest neighbors of the test sample, and the weight is Wi = 1/d (x, xi) 2 [104].

## 3. Results and Discussion

As mentioned above, the first part of this study consisted of evaluating the ability of ET to discriminate and classify different types of pesticides (categories) at a concentration of 10 ppm and in distilled water. Additional data processing techniques were applied to each of the measures acquired to identify and discriminate such pesticides with respect to distilled water.

### 3.1. Categories (Pesticides) and Distilled Water

Figure 2 shows the voltammograms recorded by the samples of the pesticides Curathane, Numetrin, Nativo, and distilled water. Some significant differences were observed between the responses generated from the carbon electrode for each category.

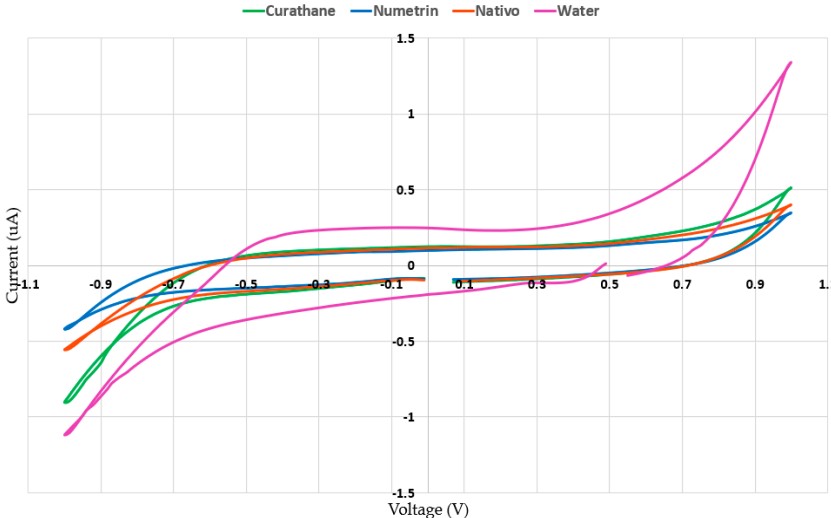

**Figure 2.** Cyclic voltammograms acquired through a bipotentiostat using the carbon sensor (C110) to detect pesticides (Curathane, Numetrin, and Nativo) and distilled water.

3.1.1. Pattern Recognition Methods

Figure 3 illustrates the result obtained by the evaluation of the carbon sensor (C110) through the PCA analysis, applying the previous normalization of the data centering and

the extraction of the static parameters: Gmax (maximum value, generated current (I)), and Gmin (minimum current value), reaching a variance of 99.8% in PC1.

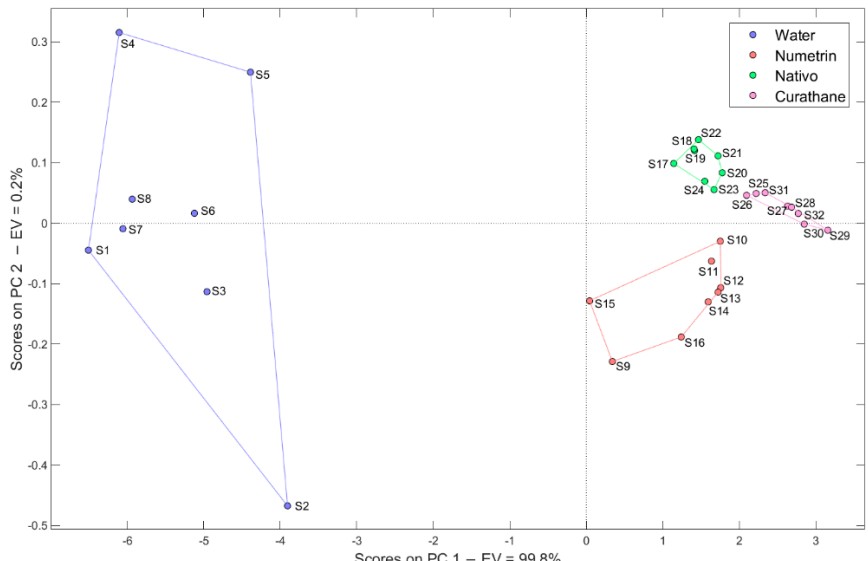

**Figure 3.** PCA plot using the carbon electrode (C110) for the discrimination of Numetrin, Nativo, and Curathane pesticides against distilled water samples.

In this main component, a greater sample representativeness was obtained; each of the categories can be discriminated, as well as clearly separating the samples of pesticides from distilled water. Although a slight dispersion of the water measurements is observed, the electrochemical process in the pesticide samples makes a more selective detection of toxic substances.

In this analysis, there were no "Outliers" or significant erroneous samples, so it was not necessary to eliminate these measurements from the data set.

Figure 4 shows a very similar behavior of the PCA acquired through the analysis of linear discriminant functions LDA in which the categories of both distilled water and pesticides in this case were classified. The information of the first two factors was calculated using the directions ("linear discriminants") that represent the axes that maximize the separation between multiple classes.

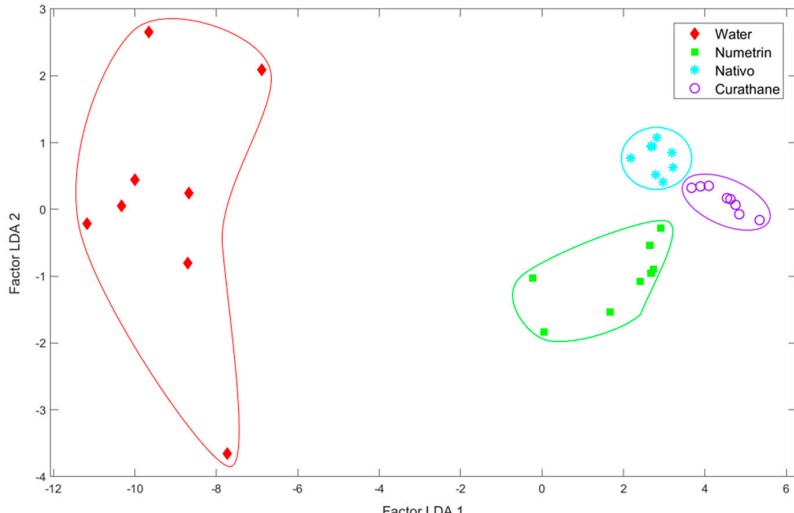

**Figure 4.** LDA plot using the carbon electrode (C110) to classify Numetrin, Nativo, and Curathane pesticides against distilled water samples.

Based on the same normalization performed on the data through data centering, it is possible to observe, as is shown on the x-axis of the graph, that Factor 1 is the component with the most significant contribution, together with Factor 2, of reduced dimensionality; therefore, the algorithm separates the categories of pesticides and distilled water into normally distributed classes.

### 3.1.2. Machine Learning

Table 1 shows the results obtained from the different data processing methods based on classifiers, which were previously described. In addition, a cross-validation method was applied to the measurements from the four categories with a k-Fold = 5. In the tests carried out with each classification method, the results were higher than 90%, obtaining 100% accuracy through the methods of naïve Bayes, SVM, and kNN.

**Table 1.** Results of the classification of pesticide types and distilled water samples using pattern recognition and machine learning methods.

| N° | Model | Accuracy |
|---|---|---|
| 1 | Linear Discriminant | 93.8% |
| 2 | Naïve Bayes | 100% |
| 3 | SVM | 100% |
| 4 | kNN | 100% |

### 3.1.3. Confusion Matrix

Figure 5 illustrates the result with the linear discriminant analysis classification method, which obtained 93.8% accuracy in the data set, where two Nativo samples were misclassified with the Curathane samples.

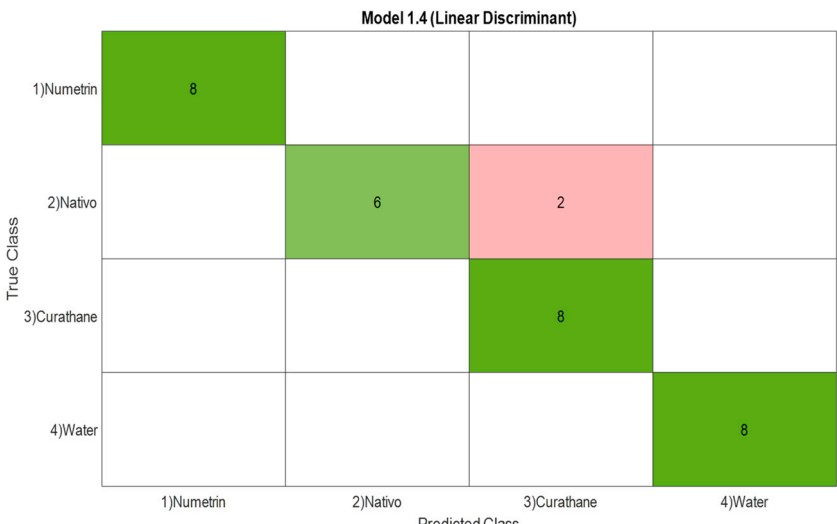

**Figure 5.** Confusion matrix obtained by applying the medium Gaussian–SVM model for classifying pesticides against distilled water samples.

Further, the confusion matrix (see Figure 6) shows the result of data classification using the SVM classifier with the medium Gaussian kernel, where a 100% success rate was obtained using the cross-validation method with k-Fold = 5. As can be seen in the graph, the classification obtained did not have any errors, and the samples were classified through the carbon sensor applying the electronic tongue.

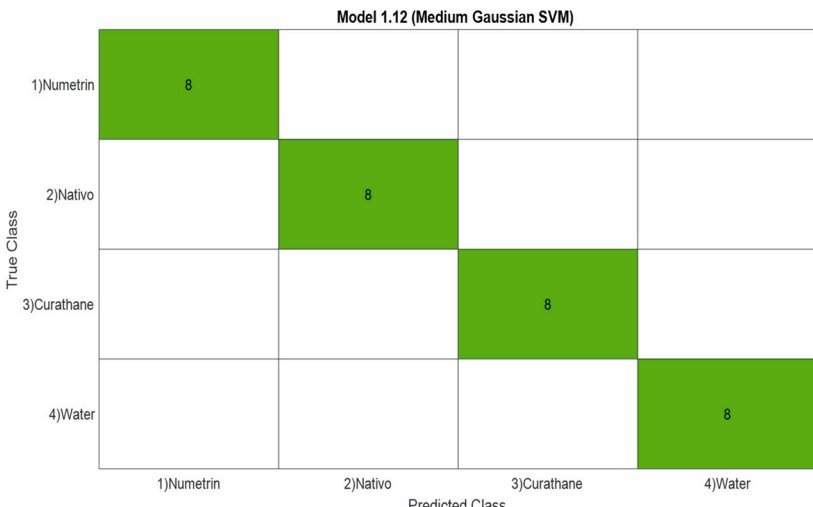

**Figure 6.** Confusion matrix using the medium Gaussian–SVM model for classifying pesticides against distilled water samples.

Figure 7 illustrates the receiver operating characteristic (ROC) and area under curve (AUC) graph, achieving 100% accuracy and 100% precision with the naïve Bayes model. The above represents the model behavior in classifying the categories based on thresholds.

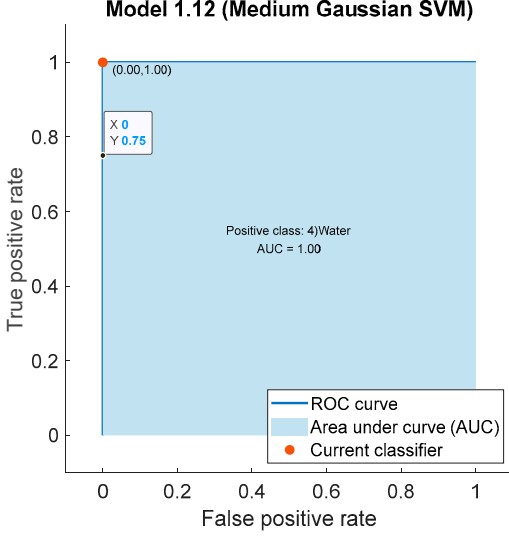

**Figure 7.** ROC-AUC curve for the medium Gaussian–SVM model for classifying different types of pesticides against distilled water samples.

### 3.2. Concentrations of Pesticides and Water

3.2.1. Curathane and Water Concentrations

In this research, the ability of the electronic tongue to establish different concentration levels of the pesticide Curathane and distilled water was evaluated. Subsequently, the results obtained from the acquired measurements and the different data processing methods that were applied for the discrimination and classification of water samples and the different concentrations of pesticides are described.

Through the processing of the pesticide categories, implementing the concentrations (0.02 to 0.25 ppm) of Curathane, it was possible to determine by projecting the measurements in the PCA graph from the selectivity and repeatability in the measurements during this analysis. It was observed that lower concentrations increase the degree of dispersion compared with samples with higher concentrations because the electrode responds differently to chemical compound concentrations when detached from their active layer (see

Figure 8). In contrast, in the detection of compounds with higher concentrations, there is greater repeatability as the sensitivity is higher in each of the electrodes, and the recovery is more noticeable.

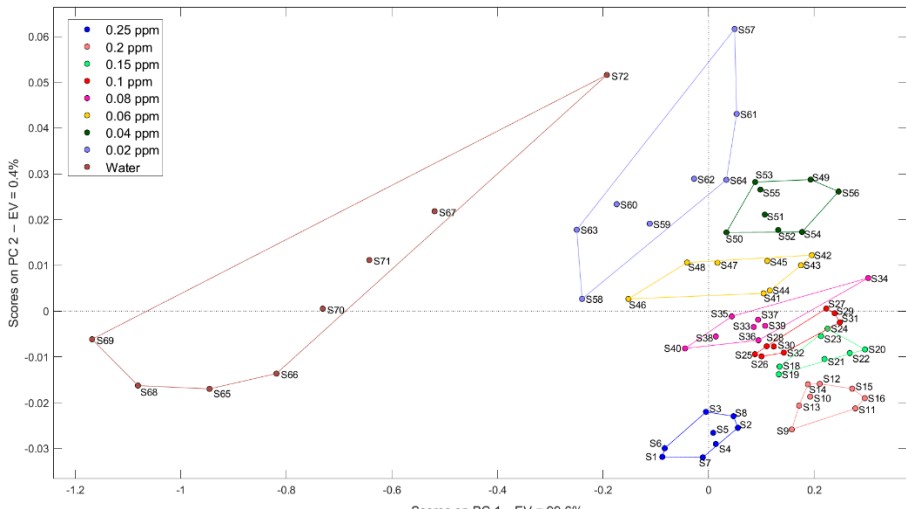

**Figure 8.** PCA plot by applying the carbon electrode (C110) to discriminate different Curathane concentrations against distilled water solution.

Similar to the PCA, Figure 9 shows a good classification of the Curathane concentrations and the distilled water samples using LDA analysis. As noticed in the PCA graph, in the extraction of factors with more than 90% variation in Factor 1, the ability of the electronic language to identify the different classes of concentrations is confirmed. In this case, the data autoscaling normalization method was applied.

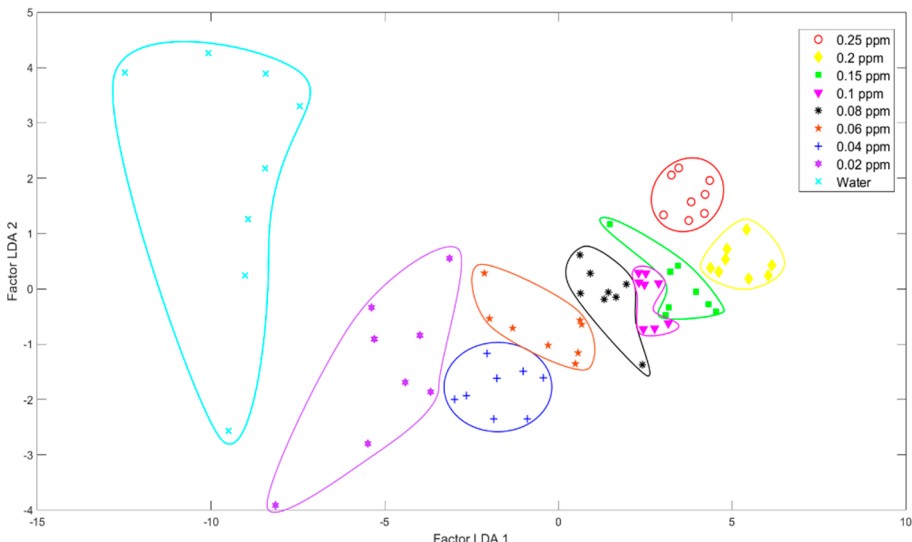

**Figure 9.** LDA plot using the carbon electrode (C110) for Curathane pesticide classification at different concentrations against distilled water.

Figure 10 illustrates the result of data classification using the kNN (kernel-weighted) classifier, which has obtained a success rate of 91.7% with k-Fold = 5. In this situation, the "scores" were used to reduce the data set dimensionality and extract relevant information from the new matrix. As seen in the graph, the classification generated six errors; there was only some overlapping in the samples of 0.15 ppm with a concentration of 0.08 ppm, 0.08 ppm with a 0.01 ppm concentration, 0.02 ppm with 0.06 ppm, and eventually, a distilled

water sample with 0.02 ppm. Despite the errors, there were few failures in classifying Curathane concentrations, obtaining a reasonable success rate.

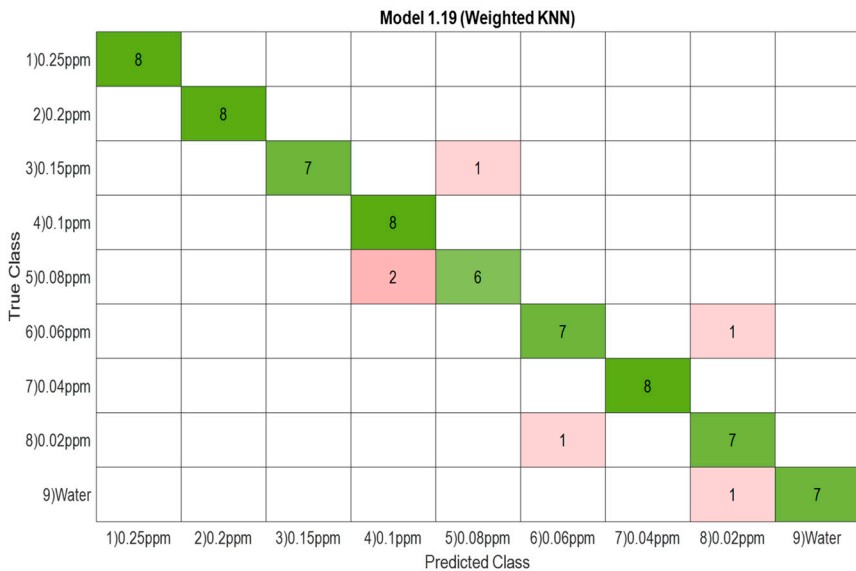

**Figure 10.** Confusion matrix for the kernel-weighted kNN model to classify Curathane pesticide at different concentrations against distilled water samples.

### 3.2.2. Nativo and Water Concentrations

For processing with the PCA method of the concentration categories of the Nativo pesticide, a similar analysis to the measurements with Curathane was carried out. It was possible to discriminate said concentrations with adequate repeatability and selectivity.

In Figure 11, a slight dispersion of the distilled water is noticed, but despite this, it is possible to identify the concentrations from lower to higher concentration. In addition, it is observed that the repeatability is better in samples with higher concentrations because of the fast electrode recovery, and the chemical compounds are separated from their active layer more efficiently.

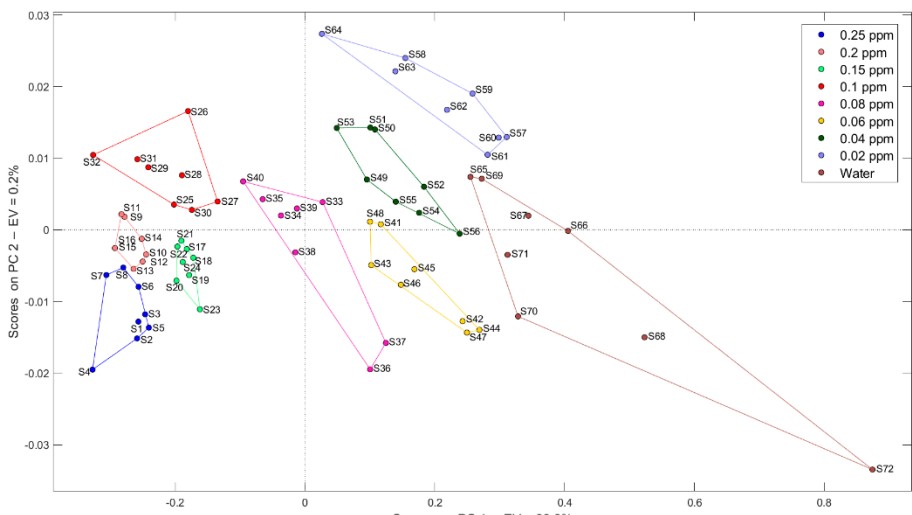

**Figure 11.** PCA plot using the carbon electrode (C110) to discriminate different Nativo pesticide concentrations against a distilled water solution.

In Figure 12, good identification of the Nativo concentrations and distilled water samples is established. As observed in the LDA graph, the information on the factors gave

a good classification with the carbon sensor (C110) of the ET for the classification of the Nativo pesticide concentrations.

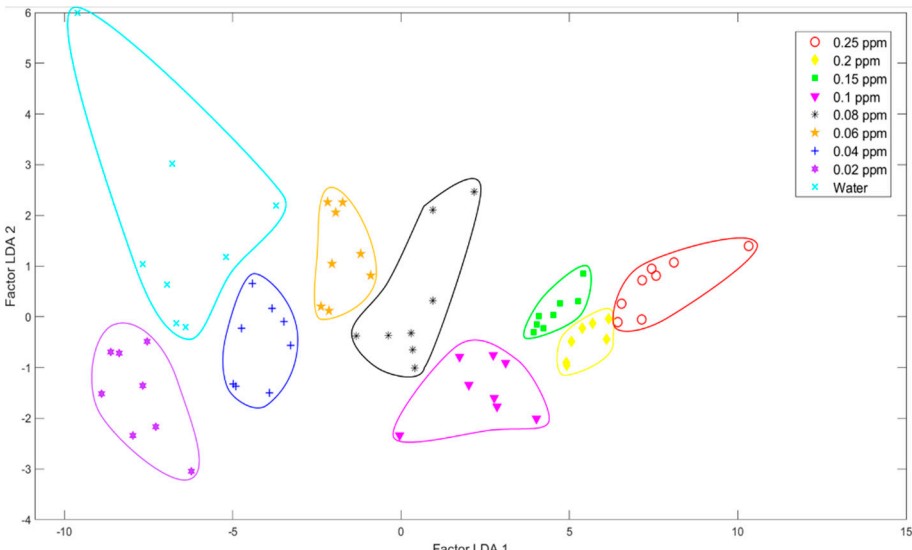

**Figure 12.** LDA plot by employing the carbon electrode (C110) for the Nativo pesticide classification at different concentrations.

Figure 13 illustrates the result of data classification using the kNN (preset: fine) classifier, which obtained a success rate of 93.1% with k-Fold = 5.

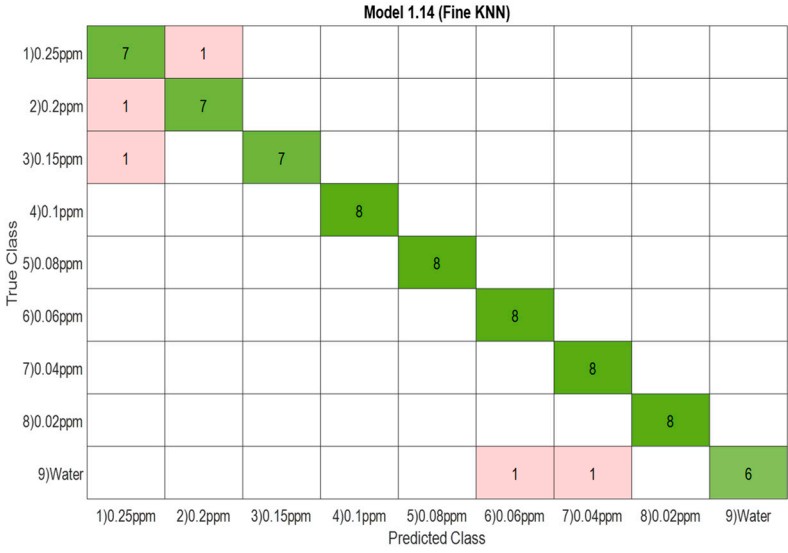

**Figure 13.** Confusion matrix for the preset-fine kNN model for classifying Nativo pesticide at different concentrations against distilled water samples.

In these tests, the classification generated five errors through the following concentrations: two errors in the concentrations 0.25 and 0.2 ppm, 0.15 with 0.25 ppm, and distilled water with the concentration of 0.04 and 0.06 ppm, respectively.

### 3.2.3. Numetrin and Water Concentrations

For the analysis of the Numetrin and distilled water samples, the PCA algorithm was applied with a previous data scaling to obtain a data variance of 100% in the sum of the two PCs. Figure 14 shows the selectivity of the Numetrin pesticide samples compared to distilled water in which the measurements with higher concentrations of the pesticide

establish greater repeatability in the detection of the compound. As mentioned above, it was observed that lower concentrations increase the degree of dispersion compared with samples with higher Numetrin concentrations because the electrode responds differently to chemical compound concentrations.

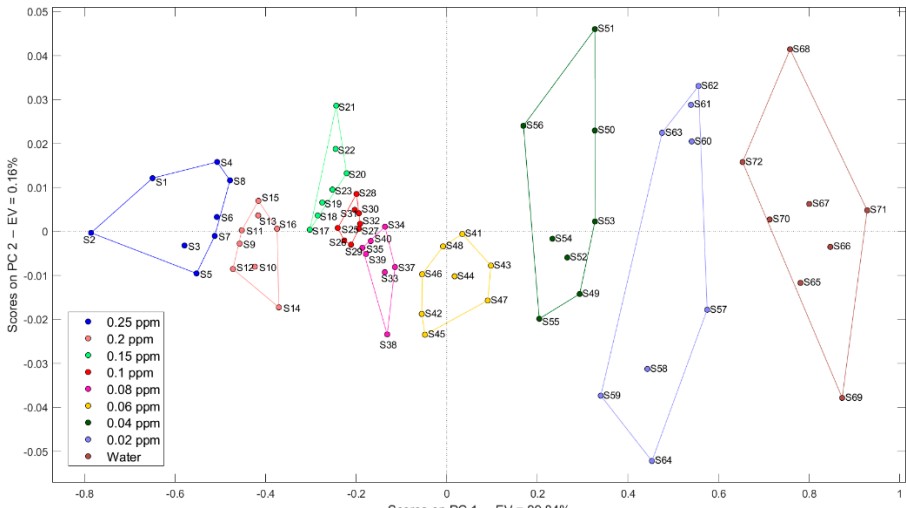

**Figure 14.** PCA plot using the carbon electrode (C110) to discriminate Numetrin pesticide concentrations in a distilled water solution.

Figure 15 represents the LDA graph, where a similar response to PCA is illustrated, as the first factors allow the Numetrin pesticides and distilled water to be correctly separated into normally distributed categories. However, some overlapping of the minor concentrations obtained with the Nativo pesticide are observed, where it is shown that the LDA with values of the normalized electrode current is an excellent strategy to be used in classification tasks.

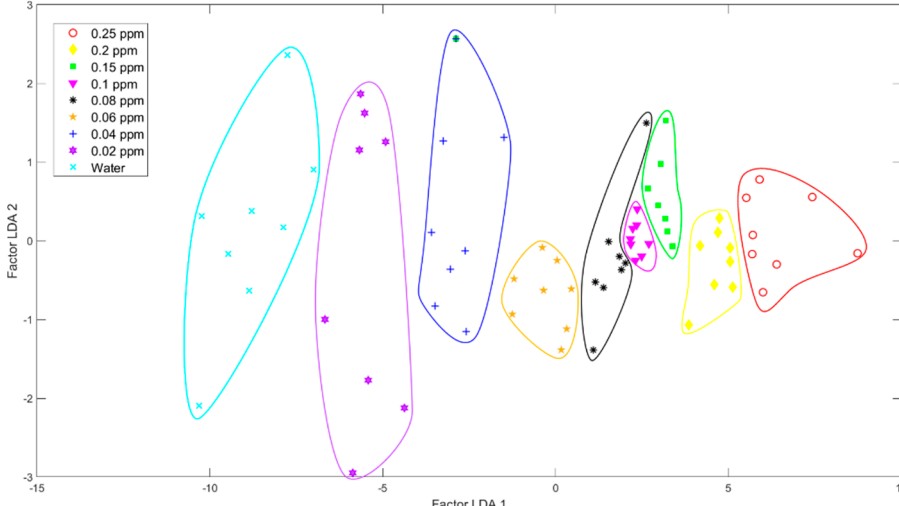

**Figure 15.** LDA plot using the carbon electrode (C110) to classify Numetrin pesticide at different concentrations against distilled water.

Figure 16 illustrates the naïve Bayes and SVM model with cubic kernel behavior, which has obtained a classification correctness rate of 90.3% in the measurements with Numetrin concentrations with distilled water samples and cross-validation technique, with k-Fold = 5. In addition, seven errors happened in the classification, where the misclassified samples were the following: 0.2 ppm was confused with the concentrations 0.15 and 0.1

ppm, 0.15 ppm with the concentration 0.04 ppm, 0.08 with 0.15 ppm, 0.04 with 0.15 ppm, the concentration 0.02 with 0.04 ppm, and water with 0.02 ppm. This classification matched the results obtained with pattern recognition techniques, as approximations of similar concentrations can be seen in both the predicted and true values.

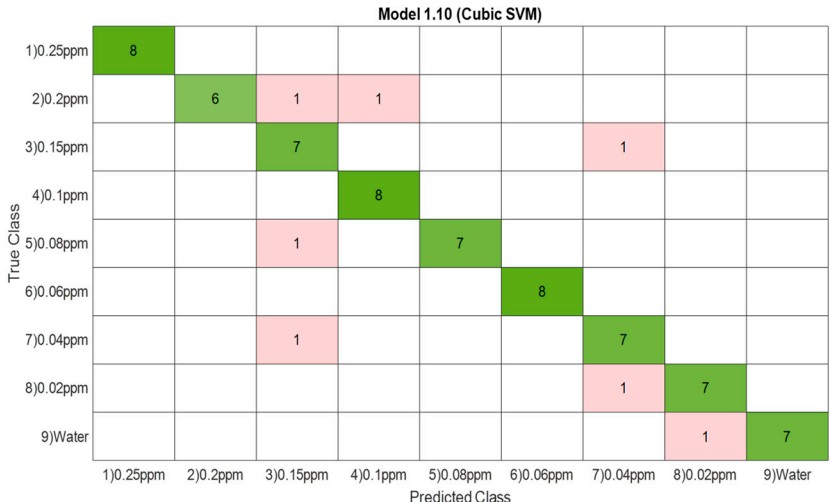

**Figure 16.** Confusion matrix for the SVM model with cubic kernel for classifying the pesticide Numetrin at different concentrations against distilled water samples.

Table 2 shows the results of the different methods used to classify each pesticide concentration and the distilled water. The results were promising, as it was possible to classify the different categories with above 90% accuracy.

**Table 2.** Classification of pesticide types and distilled water samples using machine learning methods.

| Model | Accuracy | | |
|---|---|---|---|
| | Curathane + $H_2O$ | Nativo + $H_2O$ | Numetrin + $H_2O$ |
| Linear Discriminant | 76.4% | 83.3% | 83.3% |
| Naïve Bayes | 81.9% | 77.8% | 90.3% |
| SVM | 86.1% | 93.1% | 90.3% |
| kNN | 91.7% | 93.1% | 68.1% |

In the studies found in the literature, it has been possible to demonstrate the use of electronic tongues for detecting pesticide solutions and their mixtures in micromolar and nanomolar contraction ranges. It should be noted that these studies have been carried out with sensors that have been modified with enzymes, such as electrice el (EE), genetically modified *Drosophila melanogaster* (B131), and electrice el co-immobilized with *Drosophila melanogaster* (BH), to detect different pesticides, such as paraoxon, dicorlvos, and carbofuran, and quantify the concentration of the three pesticides using machine learning methods [93]. Another electrode used was based on nanocomposites that were prepared by the reduction of graphene oxide in the presence of conductive polymers (PEDOT:PSS and polypyrrole) and gold nanoparticles (AuNP), which were deposited by dropcasting on interdigitated gold electrodes to detect organophosphorus pesticides, individual malathion (A), cadusafos (B), and concentrations in nanomolar ranges. For the data discrimination, they used the PCA technique in which it is observed that all the samples are clearly discriminated, presenting no overlap between different groups (buffer, 0.1, 0.5, 1.0, and 5.0 n mol $L^{-1}$) and a proximity of triplicate points from the same sample, which indicates a high reproducibility of the measurements. The high PC1 values obtained (PC1 represents

92.77 and 97.01% of the variance of the data for malathion and cadusafos, respectively), indicate the data correlation [94].

Therefore, the advantage of our study in comparison with previous studies is focused on the use of screen-printed electrodes which allow to small-sized, robust, cheap, mass-produced flat electrochemical cells to be easily obtained, and they can be reused in order to make more tests in a simpler and faster manner. In the same way, as it is evidenced in the research, it is the first study to discriminate and classify Curathane, Numetrin, and Nativo pesticides present in the water and samples of low concentrations of each of these, achieving approximately the admissible limit of pesticides in the water contemplated in Decree 475/1998 (Colombia).

## 4. Conclusions

The results obtained in this pilot study demonstrated the ability of the electronic tongue to discriminate and classify Curathane, Numetrin, and Nativo pesticides present in water, which are harmful to health and the environment owing to their level of toxicity and bioaccumulation. It is essential to highlight that the selection of these pesticides is given by the inquiry made to the regional farmers,

The screen-printed carbon electrode (C110) employed for the development of the sensory system proved to be stable during the different experimental tests, as it was not necessary to neither remove measurements due to "Outliers" or drifts that could have affected the measurements of each of the compounds.

It is important to note that through the data processing methods of both pattern recognition and machine learning, good results were achieved, only adjusting the configuration and parameters of algorithms, such as the SVM "kernel", with new dimensions, with which they found different hyperplanes to separate the three categories of pesticides and water. Hence, the percentage of success rate in the classification was 100% with the SVM, naïve Bayes, and kNN algorithms to classify concentrations of 10 ppm of each pesticide. Likewise, with each of the previous supervised learning methods, it was also possible to identify traces of pesticides at different concentrations from 0.25 to 0.02 ppm, obtaining a 90% success rate. The above indicates that it is possible to implement the electronic tongue with carbon electrodes to detect low concentrations of pesticides, as its response is close to the permissible levels in water, which is 0.01 ppm according to Decree 475 of 1998.

In this investigation, an electronic nose was not used, but rather an electronic tongue composed of commercial screen printing electrodes was applied. Regarding the limit of detection (LOD), in our study, it was 0.02 ppm compared to biosensors, which is lower, as they are more specific for the contaminants to be detected.

Last but not least, this innovative and simple technology could be used for water quality control, where it might be applied in specialized laboratories as a support tool due to its selectivity, speed, and low cost, as well as in an "on-site" analysis along with improving the time to obtain results and take corrective measures.

In addition, due to the versatility of electronic tongues (ETs), they can be applied in different sectors. Therefore, some studies have shown that the largest applications are focused on water analysis (microorganisms, heavy metals, quality, nutrients, etc.) [65], agricultural analysis (evaluation of inorganic ions, soil type and fertility, pesticides) [105], medical analysis by studying blood, urine, tears, sweat, and saliva, which are important biological fluids that provide information about possible health problems of a patient (cancer, urinary diseases, creatinine levels, urea, etc.) [106–108], and food quality (microorganisms, mycotoxins, maturity, acidity, heavy metals, pesticides) [71,109,110].

Several biosensor interface materials have been applied: nanomaterials (nanoparticles, quantum dots, nanowires, carbon nanotubes, dendrimers), transducers (electrochemical, electronic, optical, gravimetric, and acoustic), bioreceptors (enzymes, antibodies, and aptamers), and cells [111–113]. The selection of materials and manufacturing techniques is crucial and will depend on the application that will be given to them; for example, biosensors based on enzymes, DNA, and antibodies have demonstrated their high selec-

tivity towards specific toxic substances and, in many cases, the ability to achieve LOD below regulatory limits, making them candidates for use as fast, reliable, and cost-effective analytical techniques to support the detection of chemical contaminants (toxins, pesticides, drug residues veterinarians, and heavy metals) in different matrices (water, soil, air, food, etc.). However, the biosensors can only be used for a specific compound; one could think of developing a portable device that detects multiple contaminants in food [114,115].

**Author Contributions:** Conceptualization, J.K.C.G.; methodology, C.M.D.A.; software, C.M.D.A.; validation, Y.A.N.P., D.D.C.N., and J.K.C.G.; formal analysis, C.M.D.A. and J.K.C.G.; investigation, J.K.C.G. and D.D.C.N.; data curation, C.M.D.A.; writing original draft preparation, J.K.C.G. and Y.A.N.P., writing—review and editing, Y.A.N.P. and J.K.C.G.; visualization, Y.A.N.P.; supervision, J.K.C.G. and C.M.D.A. All authors have read and agreed to the published version of the manuscript.

**Funding:** This research received no external funding.

**Institutional Review Board Statement:** The study did not require ethical approval.

**Informed Consent Statement:** Not applicable.

**Data Availability Statement:** The data can be requested to the authors by mail.

**Acknowledgments:** The authors thank the Multisensory Systems and Pattern Recognition Research Group of Pamplona University (Pamplona, Colombia) for their support in this study.

**Conflicts of Interest:** The authors declare no conflict of interest.

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
