# Peer review of "Detection of Pesticides in Water through an Electronic Tongue and Data Processing Methods"

_water, doi:10.3390/w15040624_

Round 1

Reviewer 1 Report

Gomez et al have studied the implementation of an electronic tongue approach consisted of carbon 10 screen-printed electrodes that could detect and discriminate several pesticides including Curathane, Numetrin, and Nativo from water; with a 100% success rate in classification of compounds.

 The study is interesting and provide important findings for researchers working on pesticide analysis.

Few comments to improve the manuscript quality are as below:

1.     Please provide figures of high resolution with increased font size.

2.     The caption for figures needs to improved.

3.     Current captions for each figure are too short and difficult to understand the content of each figure.

4.     Authors could briefly introduce novel DNA-aptameric biosensors developed for discrimination and classification toxic chemicals such as pesticides, https://doi.org/10.1016/j.teac.2022.e00184; and may discuss more studies by these authors on fipronil, malathion, diazinon, and fenitrothion.

5.     Compare these eNOSE approach with aptameric sensors using fluorescence LOD, spectral analysis of pesticides, etc.

6.     Difficult to understand the figures.

7.     Compare the performance (LOD/Kd) of various sensors (enose/biosensors) with target specificity.

8.     Provide perspective and future directions for toxins and hazardous chemicals sensing.

9.     Overall, the article is interesting and have significance for pesticide analysis, however, it must be carefully revised as per above suggestions.

Author Response

Response to Reviewer 1 Comments

Dear Reviewer,

On behalf of all coauthors, we want to thank your comments in order to improve our paper. So, we are attaching our revised manuscript entitled “Detection of pesticides in water through an electronic tongue and data processing methods” which we are resubmitting for publication in Water.

We appreciate your important remarks to improve the manuscript.

-------------------------------------------------------------------------------------------------------------------

Responses to the Reviewer

  1. Please provide figures of high resolution with increased font size.

Thank you for this remark. The figure's font size was increased, and the resolution improved.

  1. The caption for figures needs to improved.

The captions for figures were improved as we did a better explanation.

  1. Current captions for each figure are too short and difficult to understand the content of each figure.

As we said above, we have done a good explanation of each figure. furthermore, in text there are more information regarding to each figure.

  1. Authors could briefly introduce novel DNA-aptameric biosensors developed for discrimination and classification toxic chemicals such as pesticides, https://doi.org/10.1016/j.teac.2022.e00184; and may discuss more studies by these authors on fipronil, malathion, diazinon, and fenitrothion.

The following references were included.

Lines: 148:156

Currently, In the review reported by Ulhas Sopanrao et al., on the nucleic acid-based aptamer sensors (single-stranded ribonucleic acids (ssRNA) or single-stranded deoxyribonucleic acids (ssDNA)) for the molecular diagnosis of toxic chemicals in food, water, human fluids and the environment[94], reports a novel aptamer-based disposable, flexible, and screen-printed electrochemical sensor (Aptasensor) for the rapid detection of chlorpyrifos (CPF). The proposed sensor showed good CPF detection capability with a low detection limit (0.097 ng/mL) and a broad linear detection range (1 to 1x105 ng/mL). The sensors also obtained good selectivity against the most common interfer-ing agents (dichlorvos, malathion, carbofuran, deltamethrin, and metamitron)[95].

  1. Inam, A.K.M.S., Angeli, M.A.C.; Douaki, A.; Shkodra, B.; Lugli, P.; Petti, L.; An Aptasensor Based on a Flexible Screen-Printed Silver Electrode for the Rapid Detection of Chlorpyrifos. Sensors 2022, 22, 7, 2754. doi: 10.3390/S22072754/S1.

Lines: 157:162

In another study, a DNA Aptamer sensor was developed to detect fenitrothion (insecticide belonging to the family of organophosphate pesticides) with a detection limit of 14 nM (3.88 ppb). Aptamer-based biosensors are often used as an alternative that provides a fast and easy way to recognize contamination from various sources. They also have the advantage of detecting low concentrations that could not be de-tected by traditional Chromatographic assays [96].

  1. Xu, Y.; Cheng, N.; Luo, Y.; Huang, K.; Chang, Q.; Panga, G.; Xu, W. An Exo III-assisted catalytic hairpin assembly-based self-fluorescence aptasensor for pesticide detection. Sens Actuators B Chem 2022, 358, 131441. doi: 10.1016/J.SNB.2022.131441.

5. Compare these eNOSE approach with aptameric sensors using fluorescence LOD, spectral analysis of pesticides, etc.

Lines: 530-533

In this investigation, an electronic nose was not used, but rather an electronic tongue composed of commercial screen printing electrodes was applied. Regarding the limit of detection (LOD), in our study it was 0.02 ppm compared to biosensors, which is lower since they are more specific for the contaminants to be detected.

  1. Difficult to understand the figures.

The figures have been improved and the explanation of each of them can be found previously the corresponding figure.

  1. Compare the performance (LOD/Kd) of various sensors (enose/biosensors) with target specificity.

It should clarify the dissociation constant (Kd) for the formation of the aptamer-object complex was not determined in this study as the biosensors were not developed, but commercial sensors were applied. As mentioned above, during the experiments was not used Electronic Nose, but in future investigations would be necessary to include other technologies to compare the previous results.    

  1. Provide perspective and future directions for toxins and hazardous chemicals sensing.

Lines: 538-557

In addition, due to the versatility of electronic tongues (ET's), they can be applied in different sectors. Therefore, some studies have shown that the largest applications are focused on water analysis (microorganisms, heavy metals, quality, nutrients, etc.)[107], agricultural analysis (evaluation of inorganic ions, soil type and fertility, pesticides)[108], medical analysis by studying the blood, urine, tears, sweat, and saliva, which are important biological fluids that provide information about possible health problems of a patient (cancer, urinary diseases, creatinine levels, urea, etc) [109]–[111] and food quality (microorganisms, mycotoxins, maturity, acidity, heavy metals, pesti-cides [112]–[114].

Several biosensor interface materials have been applied: nanomaterials (nanopar-ticles, quantum dots, nanowires, carbon nanotubes, dendrimers), transducers (electro-chemical, electronic, optical, gravimetric, and acoustic), bioreceptors (enzymes, anti-bodies, aptamers) and cells [115]–[117]. The selection of materials and manufacturing techniques is crucial and will depend on the application that will be given to them; for example, biosensors based on enzymes, DNA, and antibodies have demonstrated their high selectivity towards specific toxic substances and, in many cases, the ability to achieve LOD below regulatory limits, making them candidates for use as fast, reliable, and cost-effective analytical techniques to support in the detection of chemical con-taminants (toxins, pesticides, drug residues veterinarians, heavy metals) in different matrices (water, soil, air, food, etc.). However, the biosensors can only be used for a specific compound, one could think of developing a portable device that detects multi-ple contaminants in food [118], [119].

  1. Overall, the article is interesting and have significance for pesticide analysis, however, it must be carefully revised as per above suggestions

We have addressed all remarks that you kindly suggested.

Reviewer 2 Report

The authors address a research topic with high practical importance and definite benefits for human health and the environment. The research is extensive and the manuscript is structured and edited appropriately. There are also some typos that require correction, they will be indicated at the end. The processing of data obtained experimentally with the involvement of artificial intelligence, in particular Machine Learning, is an interesting and reliable approach.

Introduction is skillfully developed, the importance of the study is pointed out, the disadvantages of conventional pesticide detection methods are analyzed, also the level of implementation of ETs in detection techniques with application in various fields. The level of novelty and originality is satisfactory; the aim of the research is emphasized as well.

Research methodology is structured in distinct sections and is rich in relevant information for the smooth understanding of the study. Data processing methods are described accordingly. The results are analyzed in detail and are accompanied by representative graphic images, which, however, require better clarity. Discussion section is rather brief; in my opinion it presents too general information. I propose merging sections 3 and 4 or just expand section 4 with key discussions accompanying the results presented in section 3.

Overall, the manuscript is very complex, useful for the scientific sector, especially for the fields of medicine, environmental protection, agriculture and biotechnologies. Nevertheless, to be published it requires some minor improvements:

-          The first sentence (lines 158-160) is not clear, please rephrase.

-          Details of the pesticides used (producer, supplier, quality) should be provided.

-          Please check and correct the text carefully, there are remaining words typed in the native language (e g. Lines 177, Table 2).

-          The quality of the graphic images must be improved in some cases: Fig. 3, 8, 9, 11 and 14.

-           Should be Machine Learning, not Learning Machines (line 322).

-          Please use Italics for the specie Latin name (line 463). Also, more care to small typos, eg. capitalize Decree in line 484.

Author Response

Response to Reviewer 2 Comments

Dear Reviewer,

On behalf of all coauthors, we want to thank your comments in order to improve our paper. So, we are attaching our revised manuscript entitled “Detection of pesticides in water through an electronic tongue and data processing methods” which we are resubmitting for publication in Water.

We appreciate your important remarks to improve the manuscript.

-------------------------------------------------------------------------------------------------------------------

The authors address a research topic with high practical importance and definite benefits for human health and the environment. The research is extensive and the manuscript is structured and edited appropriately. There are also some typos that require correction, they will be indicated at the end. The processing of data obtained experimentally with the involvement of artificial intelligence, in particular Machine Learning, is an interesting and reliable approach.

Introduction is skillfully developed, the importance of the study is pointed out, the disadvantages of conventional pesticide detection methods are analyzed, also the level of implementation of ETs in detection techniques with application in various fields. The level of novelty and originality is satisfactory; the aim of the research is emphasized as well.

Research methodology is structured in distinct sections and is rich in relevant information for the smooth understanding of the study. Data processing methods are described accordingly. The results are analyzed in detail and are accompanied by representative graphic images, which, however, require better clarity.

-------------------------------------------------------------------------------------------------------------------

1) Discussion section is rather brief; in my opinion it presents too general information. I propose merging sections 3 and 4 or just expand section 4 with key discussions accompanying the results presented in section 3.

Lines: 306-507

We have merged sections 3 and 4 to include results and discussion in the same location or paragraph.

2) The first sentence (lines 158-160) is not clear, please rephrase.

Lines: 174-176

The selection criteria of the pesticides carried out in this study was done with the collaboration of experts in this area, which was conducted in Pamplona, Norte de Santander, Colombia.

3)    Details of the pesticides used (producer, supplier, quality) should be provided.

Lines: 183-192

The pesticides were purchased through a local supplier called “Septima Distribuciones Agropecuarias S.A.S” located in the city of Pamplona - Norte de Santander (Colom-bia). Curathane is distributed and manufactured by AgroSciences de Colombia S.A, with national registration number of ICA No. 2145; Nativo is distributed and manu-factured by Bayer S.A company, where its national registration number of the Colom-bian Agricultural Institute (ICA) is 429, and Numetrin is distributed by Nufarm Co-lombia S.A, with the national registration number of ICA No. 2074 which is manufac-tured by Gharda Chemicals Limited from India. It should be clarified that these pesti-cides are of high quality and widely used in Colombian regions.

4)  Please check and correct the text carefully, there are remaining words typed in the native language (e g. Lines 177, Table 2).

Lines: 479-480

      The text was corrected.  

 5)   The quality of the graphic images must be improved in some cases: Fig. 3, 8, 9, 11 and 14.

      Figures 3,8,9,11, and 14 have been improved (Resolution and font size)

 6)   Should be Machine Learning, not Learning Machines (line 322).

Line: 347

      This sentence was corrected

7)   Please use Italics for the specie Latin name (line 463). Also, more care to small typos, eg. capitalize Decree in line 484.

Line: 486 and 507

      The specie Latin name was changed to Italic letter, and the word decree was written in capital letter.

Round 2

Reviewer 1 Report

 Authors have revised the contents as per suggestions.